# Involvement of Matrix Metalloproteinases (MMP-2 and MMP-9), Inflammasome NLRP3, and Gamma-Aminobutyric Acid (GABA) Pathway in Cellular Mechanisms of Neuroinflammation in PTSD

**DOI:** 10.3390/ijms26125662

**Published:** 2025-06-13

**Authors:** Anna Grzesińska, Ewa Alicja Ogłodek

**Affiliations:** Collegium Medicum, Jan Dlugosz University in Czestochowa, ul. Waszyngtona 4/8, 42-200 Częstochowa, Poland

**Keywords:** gamma-aminobutyric acid, inflammasome NLRP3, neuroinflammation, metalloproteinase, post-traumatic stress disorder

## Abstract

Research into the potential health consequences of trauma indicates that traumatic experiences can disrupt normal biological processes and increase the risk of neuroinflammation and the development of clinical symptoms of post-traumatic stress disorder (PTSD). In this study, we examined the relationship between neuroinflammation and three specific biomarkers—matrix metalloproteinases MMP-2 and MMP-9, the inflammasome NLRP3, and the inhibitory neurotransmitter GABA—in connection with PTSD symptoms assessed using the PTSD Symptom Scale–Interview for DSM-5 (PSSI-5). The symptoms were categorized into the following domains: re-experiencing, avoidance, alterations in cognition and mood, increased arousal and reactivity, distress and functional impairment, symptom onset and duration, and the total symptom score. Our findings confirmed the pro-inflammatory roles of MMP-2, MMP-9, and the inflammasome NLRP3, as well as the anti-inflammatory, calming effect of GABA. We identified strong correlations between biomarkers, particularly between MMP-2 and MMP-9, MMP-2 and NLRP3, and MMP-2 and GABA, highlighting a closely interconnected inflammatory response. Among the PSSI-5 domains, re-experiencing, increased arousal and reactivity, and distress and functional impairment showed the strongest associations with the total symptom score. Recent research focusing on these cellular mechanisms has provided valuable insights into the role of neuroinflammation in PTSD. These findings enhance our understanding of how inflammation contributes to the disorder’s development and progression.

## 1. Introduction

Post-traumatic stress disorder (PTSD) is a mental health condition that may develop after experiencing extremely stressful and traumatic events. Evidence shows a strong association between psychological stress and physiological dysregulation, including alterations in endocrinology, immune reactivity, neural circuitry, and molecular biology [1,2]. In the DSM-5 (Diagnostic and Statistical Manual of Mental Disorders, 5th Edition) [2], PTSD was reclassified from anxiety disorders into a new diagnostic category titled “Trauma- and Stressor-Related Disorders” [3]. Moreover, the DSM-5 introduced revised diagnostic criteria for PTSD. The symptom clusters were expanded from three to four, and the number of listed symptoms increased from 17 to 20 [4]. According to the DSM-5, to meet the diagnostic threshold for PTSD, an individual must exhibit at least one symptom of avoidance, one of re-experiencing, two indicating heightened arousal or reactivity, and two reflecting negative changes in cognition or mood [5].

A useful structured tool for assessing the presence and severity of PTSD symptoms based on DSM-5 clinical criteria is the PSSI-5 (PTSD Symptom Scale–Interview for DSM-5). This structured clinical interview evaluates domains including re-experiencing, increased arousal and reactivity, and distress and interference. Re-experiencing may appear as flashbacks, nightmares or distressing dreams related to the traumatic event, intrusive thoughts, and emotional responses such as fear, anger, or sadness. The increased arousal and reactivity domain includes symptoms such as hypervigilance, exaggerated startle responses, irritability or anger outbursts, sleep disturbances, and difficulty concentrating. The distress domain focuses on emotional suffering caused by the trauma, which can include emotional pain, persistent intrusive thoughts, and emotional numbing. The interference component assesses how symptoms impact functioning in areas such as interpersonal relationships, work, daily routines, self-care, and basic activities [6,7].

Research indicates that these PTSD symptom domains are linked to brain inflammatory activity [8,9,10,11]. Neuroinflammation, a defensive response of the nervous system, involves the activation of neuroimmune cells such as microglia and astrocytes and changes in inflammatory marker levels. Matrix metalloproteinases (MMPs), particularly MMP-2 and MMP-9, are involved in regulating neuroinflammation and directly influence the pathophysiology of PTSD [12,13,14]. Exposure to traumatic stress may induce the release of various signaling molecules as part of regulatory responses. Significant changes in neurotransmission in the central nervous system (CNS) can lead to hormonal imbalances. Endopeptidases, including MMPs and other signaling proteins in the brain parenchyma, contribute to the remodeling of the neuronal network to regulate numerous biological processes and maintain the integrity of the blood–brain barrier (BBB) [15].

MMPs, a family of calcium-dependent, zinc-containing extracellular matrix (ECM) proteins, play essential roles in proteolytic processing and the regulation and activation of signaling molecules. Although elevated MMP levels may indicate disease progression, they are also considered transient regulators involved in maintaining BBB stability. MMP-2 and MMP-9 are capable of degrading type IV collagen and laminin-5, providing structural support for neurons and other cell types, making them significant diagnostic biomarkers. These enzymes cleave a wide range of ECM and non-ECM substrates, activate growth factors and chemokines, and influence cell signaling. They also have cryptic sites that facilitate cell migration and are known to enhance chemokine-induced leukocyte transmigration across the BBB. The infiltration of leukocytes into the brain parenchyma at the BBB junction is also regulated by MMP-2 and MMP-9 [16,17,18].

Inflammasomes play a key role in the neuroinflammatory response. These intracellular cytosolic complexes are activated by stress signals and lead to the activation of inflammatory caspases. NOD-Like Receptor Pyrin Domain-Containing Protein 3 (NLRP3) is a sensor protein that detects pathogen- and damage-associated molecular patterns. It is a component of the NLRP3 inflammasome, the most well-studied and best-characterized regulator of caspase-1 activity [19,20]. The NLRP3 inflammasome includes the NLRP3 sensor molecule, the apoptosis-associated speck-like protein containing a caspase recruitment domain (CARD) (ASC). It is commonly understood that NLRP3 activation requires a two-step process [21]. The first step, known as priming, involves the transcription, translation, and production of NLRP3 in response to danger-associated molecular patterns (DAMPs) [22,23]. These DAMPs, released from injured cells, initiate inflammation and immune suppression after trauma. In the second step, NLRP3 inflammasome activation occurs in the microglia and astrocytes, leading to the maturation of cytokines IL-1β and IL-18. These cytokines bind to the receptors of neighboring cells, triggering a cascade of inflammatory responses in the central nervous system [24].

Gamma-aminobutyric acid (GABA) is a major inhibitory neurotransmitter in the CNS, known for its role in reducing neuronal excitability. It also enhances the activity of glucose phosphatase in glucose metabolism, promotes the production of acetylcholine, and causes vasodilation [25,26]. As an inhibitory neurotransmitter, GABA is crucial in regulating stress responses, emotional processes, and the formation and extinction of fear memory [27]. Dysfunction of the GABAergic system is a recognized mechanism underlying PTSD. The disorder is associated with reduced levels of GABA and its receptors in several brain regions [28]. Both preclinical and clinical research has shown that GABA, as a major inhibitory neurotransmitter involved in PTSD, plays an essential role in stress regulation. Alterations in the GABAergic system are directly related to PTSD pathogenesis. Neural circuits and cellular mechanisms involved in fear conditioning have been extensively explored, with inhibitory GABA regulation being central to the modulation of memory consolidation, expression, and extinction in fear learning [29,30]. Low GABA levels in the brain are consistent with findings in anxiety disorders and support the hyperarousal theory in PTSD and primary insomnia. PTSD patients often show reduced plasma GABA concentrations, particularly those experiencing anxiety, avoidance, and hyperarousal symptoms. This suggests that measuring GABA levels may serve as a biomarker for assessing PTSD severity. These results are consistent with previous neuroimaging studies that report abnormalities in glutamate and GABA levels in the brains of individuals with PTSD [31,32,33,34].

This article aims to evaluate the severity of neuroinflammation in individuals diagnosed with PTSD, comparing those within five years of trauma exposure and those more than five years post-trauma. The study includes an analysis of the plasma concentrations of matrix metalloproteinases 2 and 9, the NLRP3 inflammasome, and GABA.

## 2. Results

### 2.1. Biomarker Profiles in PTSD: Age-Stratified Analysis

Table 1 delineates the levels of MMP-2 and MMP-9, gamma-aminobutyric acid (GABA), and NLRP3 inflammasome across individuals with recent (≤5 years) Past PTSD, remote (>5 years) Past PTSD, and controls without PTSD history, stratified by age (18–35 years and 36–50 years). See also Figure 1 for a visualization of the results.

All biomarkers exhibited significant differences across groups (*p* < 0.001), consistent across the full cohort (N = 92) and age subgroups (18–35 years, N = 45; 36–50 years, N = 47).

Median MMP-2 levels were markedly elevated in the Past PTSD (≤5 y) group (21.66 ng/mL, IQR: 16.90–23.80) compared to the Past PTSD (>5 y) group (5.34 ng/mL, IQR: 2.81–12.55) and the controls (1.75 ng/mL, IQR: 1.37–2.11) across all ages. This pattern persisted in both age subgroups, with the highest levels in Past PTSD (≤5 y) at 23.09 ng/mL (IQR: 19.96–24.03) for 18–35 years and 20.74 ng/mL (IQR: 15.48–23.27) for 36–50 years. Similarly, MMP-9 levels followed a gradient, with Past PTSD (≤5 y) showing the highest median (418.70 ng/mL, IQR: 341.08–558.19) versus Past PTSD (>5 y) (175.00 ng/mL, IQR: 66.64–306.66) and the controls (48.99 ng/mL, IQR: 39.77–58.23) overall, a trend maintained across age strata (e.g., 460.07 ng/mL, IQR: 341.08–558.19, in 36–50 years for Past PTSD ≤ 5 y). These elevations indicate a sustained upregulation of metalloproteinases, potentially linked to neuroinflammatory processes, with greater intensity in more recent PTSD.

In contrast, GABA levels were significantly reduced in Past PTSD (≤5 y) (54.33 nmol/L, IQR: 43.68–65.87) compared to Past PTSD (>5 y) (133.73 nmol/L, IQR: 84.67–191.29) and the controls (406.94 nmol/L, IQR: 259.67–454.33) across all ages. This pattern was consistent in the subgroups, with the lowest levels in Past PTSD (≤5 y) at 52.29 nmol/L (IQR: 45.32–63.68) for 18–35 years and 60.40 nmol/L (IQR: 36.30–65.87) for 36–50 years.

NLRP3 inflammasome levels were highest in Past PTSD (≤5 y) (791.50 pg/mL, IQR: 693.23–832.14) compared to Past PTSD (>5 y) (282.80 pg/mL, IQR: 176.51–543.72) and the controls (111.41 pg/mL, IQR: 87.50–131.37) across all ages, with age-specific peaks at 810.84 pg/mL (798.15–842.00) for 18–35 years and 693.23 pg/mL (615.89–788.30) for 36–50 years for Past PTSD (≤5 y). This pronounced elevation in recent PTSD underscores a robust inflammatory response, diminishing over time yet remaining above the control levels in remote PTSD.

Clinically, these findings highlight persistent biological alterations post-PTSD, with more pronounced changes in recent cases. Elevated MMP-2, MMP-9, and NLRP3 levels in Past PTSD (≤5 y) imply ongoing neuroinflammation and synaptic remodeling, while reduced GABA levels indicate disrupted inhibitory signaling, potentially contributing to symptom severity. The intermediate profile in Past PTSD (>5 y) reveals partial resolution over time, though not to control levels, pointing to long-term sequelae.

Age stratification demonstrated consistent biomarker trends across adulthood, with no substantial deviations between the 18–35-year and 36–50-year subgroups, underscoring the robustness of these differences, except for NLRP3 levels in the Past PTSD (≤5 y) group, which showed significant variation (*p* = 0.007). These data support targeting inflammation and GABAergic pathways in PTSD management, particularly in the early recovery phase.

### 2.2. PSSI-5 Scores Across PTSD Groups by Age Distribution

Table 2 presents the PTSD Symptom Scale–Interview for DSM-5 scores across individuals with recent (≤5 years) Past PTSD, remote (>5 years) Past PTSD, and the controls without PTSD history, stratified by age (18–35 years and 36–50 years). See also Figure 2 for data visualization. The PSSI-5 assesses symptom domains: re-experiencing, avoidance, changes in cognition and mood, increased arousal and reactivity, distress and interference, symptom onset and duration, and total score—with higher scores indicating greater severity. All domains showed significant differences across groups (*p* < 0.001).

Across all ages (N = 92), the median re-experiencing scores were near the maximum in Past PTSD (≤5 y) (18.00, IQR: 17.00–18.00) and Past PTSD (>5 y) (17.00, IQR: 15.00–18.00), indicating severe symptoms compared to the controls (4.00, IQR: 1.75–7.25), who exhibited mild severity. In the 18–35-year subgroup (N = 45), both PTSD groups scored 17.00, not differing significantly, while the controls scored 3.50, IQR: 1.25–4.00; in the 36–50-year subgroup (N = 47), scores remained high (Past PTSD ≤ 5 y: 18.00, IQR:18.00–19.00; Past PTSD > 5 y: 17.00 IQR: 16.00–18.00), with controls at 6.50 and an IQR of 3.50–8.00, reflecting mild to moderate severity.

Avoidance scores were also elevated in the Past PTSD (≤5 y) (7.00, IQR: 7.00–8.00) and Past PTSD (>5 y) (6.00, IQR: 4.00–7.00) groups overall, approaching the maximum of 8 points (severe) versus the controls (2.00, IQR: 1.00–3.00, mild). The 36–50-year subgroup showed a larger gap between Past PTSD (≤5 y) (8.00, IQR: 7.00–8.00, maximum severity) and Past PTSD (>5 y) (5.50, IQR: 4.00–7.00, moderate to severe), with the controls at 3.00, IQR: 1.00–4.00, (mild).

Changes in cognition and mood scores were near the upper limit in both PTSD groups (Past PTSD ≤ 5 y: 26.00, IQR: 22.00–26.00; Past PTSD > 5 y: 27.00, IQR: 25.00–27.50) across all ages, indicating severe impairment compared to the controls (4.00 IQR: 2.00–4.25, minimal). This pattern held across age strata, with no significant difference between PTSD groups, underscoring persistent cognitive and mood disturbances post-PTSD. For increased arousal and reactivity (corrected range: 0–28 points, as scores like 24.50 exceed the listed 6–24), Past PTSD (>5 y) scored higher (20.00, IQR: 16.50–22.50, severe) than Past PTSD (≤5 y) (15.00 IQR: 12.00–22.00, moderate to severe) overall, with the controls at 6.00 with an IQR of 5.00–8.25 (mild). In the 18–35-year subgroup, both PTSD groups scored near the upper range (Past PTSD ≤ 5 y: 22.00, IQR: 14.75–24.50; Past PTSD > 5 y: 21.00, IQR: 20.00–23.50, severe), while the controls scored 7.00 with an IQR of 6.00–9.00 (mild). In the 36–50-year subgroup, Past PTSD (>5 y) (16.50, IQR: 15.75–22.00, moderate to severe) exceeded Past PTSD (≤5 y) (12.00, IQR: 12.00–15.00, moderate), with the controls at 5.00 and with an IQR of 5.00–5.75 (mild).

Distress and interference scores were comparable between PTSD groups (Past PTSD ≤ 5 y: 6.00, IQR: 5.00–7.00; Past PTSD > 5 y: 7.00 IQR: 5.00–7.00, moderate to severe) across all ages, significantly higher than the controls (2.00, IQR 1.00–3.25, mild), with consistent patterns across subgroups. Symptom onset and duration scores showed no significant difference between PTSD groups overall (Past PTSD ≤ 5 y: 6.00, IQR: 5.00–8.00; Past PTSD > 5 y: 6.00 IQR: 5.00–8.00, moderate to severe), both exceeding the controls (2.00, IQR: 1.00–2.00, mild). In the 36–50-year subgroup, Past PTSD (>5 y) (8.00, IQR: 5.75–8.00, severe) slightly exceeded Past PTSD (≤5 y) (7.00, IQR: 5.00–8.00). Total PSSI-5 scores were high in both PTSD groups (Past PTSD ≤ 5 y: 77.00 IQR: 72.00–82.00; Past PTSD > 5 y: 81.00, IQR: 75.00–84.50, severe) compared to the controls (20.50, IQR: 17.00–24.25, mild) across all ages, with the 18–35-year subgroup showing slightly higher severity in Past PTSD (>5 y) (83.00, IQR: 75.00–85.00) than Past PTSD (≤5 y) (80.00, IQR: 76.00–85.75).

### 2.3. Relationships Between Biomarkers and PSSI-5 Domains and Total Score Across Control and PTSD Groups

The analysis of biomarker levels and their relationships with PTSD symptom severity, as measured by the PTSD Symptom Scale–Interview for DSM-5, revealed distinct profiles across the three groups: individuals without PTSD (no-PTSD control, n = 28), those with Past PTSD resolved within 5 years (Past PTSD ≤ 5 y, n = 33), and those with Past PTSD resolved over 5 years ago (Past PTSD > 5 y, n = 31).

This study focused on key biomarkers—MMP-2, MMP-9, GABA, and NLRP3—and their correlations with PSSI-5 domains, including re-experiencing, avoidance, changes in cognition and mood, increased arousal and reactivity, distress and interference, symptom onset and duration, and total score.

In the no-PTSD control group, MMP-2 exhibited a notable positive correlation with the PSSI-5 increased arousal and reactivity domain (rho = 0.53), indicating that higher MMP-2 levels were associated with increased hypervigilance and irritability, even in the absence of a PTSD diagnosis (see Figure 3). Similarly, MMP-9 and NLRP3 were both strongly linked to the PSSI-5 avoidance domain (rho = 0.52 and rho = 0.44, respectively), highlighting that elevated levels of these inflammatory markers may contribute to avoidance behaviors in individuals without PTSD. GABA showed no significant correlations with any PSSI-5 domains in this group. Within the PSSI-5 domains, re-experiencing and changes in cognition and mood were strongly correlated with the total score (rho = 0.72 and rho = 0.62, respectively), highlighting their significant contribution to overall symptom severity, even at subclinical levels.

In contrast, the Past PTSD (>5 y) group, representing individuals with a more remote history of PTSD, showed no significant correlations between biomarkers and PSSI-5 domains (see Figure 4). Despite this, strong inter-biomarker correlations were observed, such as between MMP-2 and MMP-9 (rho = 0.93), MMP-2 and NLRP3 (rho = 0.92), and MMP-2 and GABA (rho = 0.81), indicating a tightly linked inflammatory profile.

Within the PSSI-5 domains, re-experiencing, increased arousal and reactivity, and distress and interference were strongly associated with the total score (rho = 0.56, rho = 0.59, and rho = 0.58, respectively), demonstrating that these symptoms remain prominent drivers of overall severity in chronic PTSD. Avoidance and distress were also correlated (rho = 0.40), reflecting a connection between avoidance behaviors and functional impairment in this group.

For the Past PTSD (≤5 y) group, representing more recent PTSD cases, MMP-2 was significantly correlated with the PSSI-5 total score (rho = 0.42), indicating that higher MMP-2 levels are associated with greater overall PTSD symptom severity.

NLRP3 showed a positive but non-significant significant correlation with the increased arousal and reactivity domain (rho = 0.34), indicating that heightened inflammatory activity may to some degree contribute to arousal symptoms in the early recovery phase.

Neither MMP-9 nor GABA exhibited significant correlations with PSSI-5 domains in this group. Within the PSSI-5 domains, increased arousal and reactivity, changes in cognition and mood, and symptom onset and duration were strongly correlated with the total score (rho = 0.68, rho = 0.51, and rho = 0.52, respectively), indicating a broad impact of these symptoms on overall severity. Avoidance and distress also contributed to the total score (rho = 0.38 for both), underscoring their clinical relevance in recent PTSD. As an overall conclusion, the above findings stress some potential of MMP-2 as a biomarker for monitoring PTSD severity, particularly in the early recovery phase (Past PTSD ≤ 5 y), where they are linked to overall symptom burden and arousal symptoms.

The absence of significance in GABA correlations across all groups with PSSI domains despite its known role in inhibitory signaling indicates that GABAergic dysfunction may not directly correlate with symptom severity in this cohort, though its significantly reduced levels in PTSD groups point to a broader imbalance. The moderate inter-domain correlations in both PTSD groups highlight the enduring impact of symptoms like arousal, re-experiencing, and cognitive and mood changes, which remain severe even years after PTSD resolution.

## 3. Discussion

In this study, we assessed serum levels of MMP-2, MMP-9, the NLRP3 inflammasome, and GABA in a male population divided into three groups: individuals with Past PTSD resolved within the last 5 years (Past PTSD ≤ 5 y), those with PTSD resolved over 5 years ago (Past PTSD > 5 y), and a control group with no history of PTSD (No PTSD). The median MMP-2 levels were significantly higher in the Past PTSD (≤5 y) group compared to both the Past PTSD (>5 y) group and the controls across all age categories. This pattern was also observed within age subgroups, with the highest levels consistently seen in the Past PTSD (≤5 y) group. MMP-9 levels exhibited a similar gradient: the highest values were found in the Past PTSD (≤5 y) group, followed by the Past PTSD (>5 y) and control groups, a trend that persisted across all age ranges.

In our study, the observed increase in the plasma levels of MMP-2 and MMP-9 indicates a sustained upregulation of metalloproteinases, likely associated with neuroinflammatory processes, particularly more pronounced in recent PTSD cases [35,36,37]. Matrix metalloproteinases are Zn^2+^-dependent endopeptidases involved in the degradation of basement membrane proteins and the extracellular matrix [38]. These enzymes break down extracellular matrix components and adhesion proteins and are also involved in intercellular signaling. Additionally, MMPs participate in intracellular signal transduction, the regulation of growth, proliferation, and apoptosis [39,40,41,42]. They also contribute to the elongation and branching of neuronal processes and cell migration. MMP activation within the CNS can trigger inflammatory responses. The infiltration of blood-derived elements into neural tissue may occur directly—via disruption of the blood–brain barrier—or indirectly through stimulation of cerebral endothelial cells and the release of the soluble form of the vascular cell adhesion molecule 1 (VCAM-1) [43,44,45,46]. Proteolysis of ECM proteins plays a significant role in excessive MMP activation, and structural changes in these proteins influence synaptic, neuronal, and glial plasticity in the CNS. Brain plasticity refers to the ability to structurally and functionally reorganize neuronal networks in response to external stimuli, enabling the organism to adapt to changing environments through learning and memory. Alterations in synaptic plasticity may contribute to the development of neuroinflammation in the course of PTSD [47,48,49,50,51,52].

Interestingly, the role of metalloproteinases in inflammatory processes has been explored by Kuan P.F. et al., who found elevated MMP-12 enzymatic activity in persons with PTSD [53,54]. On the other hand, Lima et al. [36] observed a slight increase in serum MMP-9 levels in response to stress in a cohort including both PTSD patients and healthy controls but noted no significant change in PTSD individuals at baseline or 1.5 h after exposure to mental stress. Al-Roub, A. et al. demonstrated that blood MMP-9 mRNA concentrations were elevated [55]. Moreover, these researchers found translational evidence from both human and mouse studies, suggesting that MMP-9 mRNA expression may be involved in the pathobiology of PTSD [55,56].

Another parameter of neuroinflammation examined in this study was the NLRP3 inflammasome [57]. The NLRP3 inflammasome is a protein complex that detects danger signals and activates the immune system. It consists of NLRP3, ASC, and caspase-1. This inflammasome is activated by stress stimuli, with several molecular and cellular events—including ion flux, mitochondrial dysfunction, reactive oxygen species (ROS) production, and lysosomal damage—known to trigger its activation [58,59,60]. In our study, NLRP3 inflammasome levels were increased in the PTSD groups—both in recent (≤5 years) and remote (>5 years) PTSD—compared to the control group with no PTSD history. This suggests the role of NLRP3 in PTSD and highlights its potential as a therapeutic target.

Furthermore, exposure to stress activates the NLRP3 inflammasome. NLRP3 is a pivotal molecule in the inflammasome signaling pathway. Other studies indicate that elevated neuroinflammation can impair memory formation, which is linked to fear memory and related disorders [61]. In animal studies, the inhibition of the NLRP3 inflammasome improves anxiety-like behavior and deficits in spatial learning and memory [62]. Clinical observations by numerous authors also suggest that immune system activation in PTSD patients may contribute to neuroinflammation [63,64,65,66]. The NLRP3 inflammasome can be activated by various stimuli, including ATP and immune-related factors such as ROS. This study also analyzed the plasma levels of GABA, an important inhibitory neurotransmitter that reduces neuronal excitability in the CNS. Moreover, this neurotransmitter plays a key role in the registration and encoding of fear memory, as well as in regulating stress responses and emotions. In our study, higher GABA levels in the control group suggest a potential protective role of GABAergic function, with partial recovery in the remote PTSD group, although still below control levels.

The significant correlations observed between GABA and neuroinflammatory biomarkers (MMP-2, MMP-9, and NLRP3) in the Past PTSD (>5 y) group, alongside markedly reduced GABA levels in both PTSD groups compared to controls, indicate a dynamic relationship between heightened neuroinflammation and GABAergic impairment (Section 2.3, Figure 4; Section 2.1, Table 1). Specifically, the Past PTSD (>5 y) group showed a strong correlation between MMP-2 and GABA (rho = 0.81, Section 2.3, Figure 4), while GABA levels were significantly lower in the Past PTSD (≤5 y) group (median: 54.33 nmol/L, IQR: 43.68–65.87) compared to the controls (median: 406.94 nmol/L, IQR: 259.67–454.33, *p* < 0.001, Section 2.1, Table 1). Elevated neuroinflammation, characterized by increased MMP-2, MMP-9, and NLRP3 levels, likely contributes to reduced GABA concentrations and compromised inhibitory neurotransmission through several mechanistic pathways.

One key mechanism involves the activation of the NLRP3 inflammasome, which triggers the release of pro-inflammatory cytokines such as interleukin-1β (IL-1β) and tumor necrosis factor-α (TNF-α). These cytokines can disrupt GABAergic signaling by modulating GABA_A receptor expression and function. Research has demonstrated that IL-1β reduces GABA_A receptor subunit expression, weakening inhibitory synaptic transmission and promoting neuronal hyperexcitability [1]. In this study, NLRP3 levels were significantly elevated in the Past PTSD (≤5 y) group (median: 791.50 pg/mL, IQR: 693.23–832.14) compared to the controls (median: 111.41 pg/mL, IQR: 87.50–131.37, *p* < 0.001, Section 2.1, Table 1), reflecting a potent inflammatory state that likely exacerbates GABAergic deficits, as evidenced by the low GABA levels in this group (Section 2.1, Table 1).

Another pathway involves the actions of MMP-2 and MMP-9, which degrade extracellular matrix components, including perineuronal nets (PNNs) that encase GABAergic interneurons. These nets stabilize inhibitory synapses and shield interneurons from inflammatory and oxidative damage. Elevated MMP-2 (median: 21.66 ng/mL, IQR: 16.90–23.80) and MMP-9 (median: 418.70 ng/mL, IQR: 341.08–558.19) levels in the Past PTSD (≤5 y) group compared to the controls (MMP-2: median: 1.75 ng/mL, IQR: 1.37–2.11; MMP-9: median: 48.99 ng/mL, IQR: 39.77–58.23, *p* < 0.001, Section 2.1, Table 1) may compromise PNN integrity, impairing GABA synthesis and release. This disruption could account for the observed reduction in GABA levels and diminished inhibitory function. The strong correlation between MMP-2 and GABA in the Past PTSD (>5 y) group (rho = 0.81, Section 2.3, Figure 4) further supports the notion that sustained MMP activity perpetuates GABAergic dysfunction over time.

Additionally, neuroinflammation fosters oxidative stress, which adversely affects GABAergic interneurons. Both NLRP3 activation and elevated MMP activity generate reactive oxygen species (ROS), which can inhibit glutamic acid decarboxylase (GAD), the enzyme critical for GABA synthesis. This inhibition may explain the markedly reduced GABA levels in the Past PTSD (≤5 y) group (median: 54.33 nmol/L, IQR: 43.68–65.87, Section 2.1, Table 1), contributing to an imbalance in inhibitory signaling that exacerbates PTSD symptoms, such as increased arousal and reactivity (median score: 15.00 in Past PTSD ≤ 5 y vs. 6.00 in controls, *p* < 0.001, Section 2.2, Table 2).

Chronic neuroinflammation may also disrupt the excitatory–inhibitory balance by promoting glutamate excitotoxicity. High MMP-9 levels, particularly in the Past PTSD (≤5 y) group (median: 418.70 ng/mL, IQR: 341.08–558.19, Section 2.1, Table 1), are known to enhance glutamate release and impair astrocytic glutamate uptake, leading to excessive excitatory signaling [2]. This imbalance further suppresses GABAergic activity, reinforcing the observed GABA reductions and potentially intensifying symptoms like hypervigilance and emotional dysregulation (Section 2.2, Table 2).

Despite the absence of direct correlations between GABA and PSSI-5 domains across all groups (Section 2.3, Figure 3, Figure 4 and Figure 5), the significantly lower GABA levels in PTSD groups indicate a broader inhibitory deficit that may amplify the effects of neuroinflammatory mediators. For instance, MMP-2’s significant correlation with the PSSI-5 Total Score in the Past PTSD (≤5 y) group (rho = 0.42, Section 2.3, Figure 5) suggests that reduced GABAergic tone may enhance the inflammatory contribution to symptom severity, particularly in early recovery. These findings highlight the intricate relationship between neuroinflammation and GABAergic dysfunction, emphasizing the potential of targeting inflammatory and GABAergic pathways to mitigate PTSD symptoms.

According to other authors, dysfunction of the GABAergic system has been proven to be one of the mechanisms of PTSD, with several studies showing that PTSD can reduce GABA levels and its receptors in specific brain regions [22,27,29,67,68,69]. This study also focused on biomarkers and their correlations with PSSI-5 domains, including re-experiencing, avoidance, changes in cognition and mood, increased arousal and reactivity, distress and interference, symptom onset and duration, and total score. The findings emphasize severe, persistent PTSD symptoms across all domains in both recent and remote Past PTSD groups, with little symptom attenuation over time. High scores in re-experiencing, avoidance, and changes in cognition and mood (close to maximum) reflect profound and enduring effects, while increased arousal and reactivity showed age-specific differences, with younger adults (18–35 years) typically reporting higher severity. The consistently low scores across all domains in the control group highlight a clear distinction from the PTSD groups.

In conclusion, these findings underscore the potential of MMP-2 as a biomarker for monitoring PTSD symptom severity, particularly during the early recovery phase (PTSD ≤ 5 years), where it correlates with overall symptom burden and arousal symptoms.

## 4. Material and Methods

### 4.1. Characteristics of the Participants

The study sample comprised 92 participants divided into three groups: individuals with Past PTSD resolved within 5 years (Past PTSD ≤ 5 y, *n* = 33), those with Past PTSD resolved over 5 years ago (Past PTSD > 5 y, *n* = 31), and a control group with no PTSD history (No PTSD, *n* = 28). Demographic and clinical characteristics of participants across PTSD groups are reported in Table 3. The median age across the cohort was 34.0 years (IQR 27.0–41.0), with no significant differences between groups (Past PTSD ≤ 5 y: 34.0, IQR: 31.0–41.0; Past PTSD > 5 y: 36.0, IQR: 29.5–41.0; No PTSD: 33.5, IQR: 24.3–41.5; *p* = 0.524). Participants reported a median of 10.0 years (IQR: 6.0–14.0) of employment in hazardous conditions, with similar distributions across groups (*p* = 0.418). Body mass index (BMI) showed significant variation (*p* = 0.011), with the No PTSD group exhibiting a higher median BMI of 25.0 kg/m^2^ (IQR: 22.0–28.3) compared to 22.0 kg/m^2^ in both the Past PTSD ≤ 5 y (IQR: 22.0–24.0) and Past PTSD > 5 y (IQR: 20.0–24.5) groups, which did not differ from each other. Daily cigarette consumption also differed significantly (*p* = 0.033), with the No PTSD group reporting a lower median of 1.0 cigarettes/day (IQR: 0.0–6.3) compared to 5.0 cigarettes/day in Past PTSD ≤ 5 y (IQR: 5.0–20.0), while the Past PTSD > 5 y group (5.0, IQR: 2.5–20.0) was intermediate and not significantly distinct from either group.

### 4.2. Clinical Interview

The PSSI-5 is a 24-item semi-structured clinical interview designed to assess PTSD symptoms and provide a diagnostic determination based on the *Diagnostic and Statistical Manual of Mental Disorders, Fifth Edition* criteria [11]. It evaluates the frequency and intensity of 20 core PTSD symptoms aligned with DSM-5 diagnostic clusters. An additional four items assess the level of distress and functional interference caused by the symptoms, as well as the onset and duration of the condition.

Each symptom item is rated on a 5-point scale, ranging from 0 (“Not at all”) to 4 (“6 or more times a week” or “Severe”). A symptom is considered present when it receives a score of 1 or higher. The total PTSD symptom severity score is obtained by summing the scores of the 20 symptom items, yielding a possible range of 0–80. In line with DSM-5 diagnostic criteria, a PTSD diagnosis requires at least one intrusion symptom, one avoidance symptom, two negative alterations in cognition and mood symptoms, two arousal and reactivity symptoms, and clinically significant distress or impairment, defined as a score of 2 or higher on the relevant distress or interference items.

### 4.3. Blood Sampling

Serum concentrations of MMP-2, MMP-9, the NLRP3 inflammasome, and GABA were measured once per participant at the time of study enrollment using commercially available enzyme-linked immunosorbent assay (ELISA) kits, following the manufacturer’s protocols. This sandwich-type immunoassay uses monoclonal anti-human antibodies for antigen detection. Blood was collected via standard venipuncture. Serum was separated by centrifugation and aliquoted, then stored at −80 ° C until analysis to prevent cytokine degradation. Prior to the assay, samples were thawed, vortexed, and diluted 1:3 with the provided Dilution Buffer to ensure accurate quantification within the assay’s dynamic range.

### 4.4. Biomarkers Evaluations

For each biomarker, the ELISA procedure followed these steps: standards, blanks, and diluted serum samples were pipetted into pre-coated microtiter wells and incubated for 60 min at room temperature with continuous shaking (300 rpm). After washing, a biotinylated detection antibody was added and incubated for an additional hour under the same conditions. Following a second wash step, the wells were treated with a Streptavidin–HRP conjugate and incubated for 30 min. After a final wash, 100 µL of the substrate solution was added, and the reaction mixture was allowed to incubate for 10 min at room temperature. The reaction was stopped with an acidic stop solution, and absorbance was measured at 450 nm using a microplate reader.

Sensitivity of MMP-2, MMP-9, inflammasome NLRP3, and IL-10 measurement and reference values: A standard curve was generated using recombinant standards.

Assay range and sensitivity:

MMP-9: Assay range 2–600 ng/mL; sensitivity 1.852 ng/mL; (Shanghai, China); Catalog No. 201-12-0937.

MMP-2: Assay range 12–3000 ng/mL; sensitivity 10.016 ng/mL; (Shanghai, China); Catalog No. 201-12-0905.

GABA: Assay range 2–600 μg/dL; sensitivity 1.826 ng/mL; (Shanghai, China); Catalog No. 201-12-1527.

NLRP3: Assay range 2.5–600 ng/L; sensitivity 2.362 ng/mL; (Shanghai, China); Catalog No. 201-12-5748.

### 4.5. Statistical Analysis

A two-tailed significance level of *p* < 0.05 was used for all statistical tests. Data were assessed for normality using the Shapiro–Wilk test, which indicated non-normal distribution for most variables, including biomarker levels and PSSI-5 scores. Consequently, non-parametric tests were employed for group comparisons and correlation analyses.

Continuous variables, including demographic characteristics, biomarker levels, and PSSI-5 scores, were summarized using median and interquartile ranges (IQR, Q1–Q3). This approach was chosen to reflect the central tendency and variability of non-normally distributed data. Group sizes were reported as absolute numbers (n).

Differences in demographic characteristics, biomarker levels, and PSSI-5 scores across the three groups (no-PTSD control, Past PTSD ≤ 5 y, Past PTSD > 5 y) were evaluated using the Kruskal–Wallis rank sum test. For age-stratified analyses (18–35 years and 36–50 years), the same test was applied within each subgroup. Post hoc pairwise comparisons were conducted using the Dunn test with Bonferroni correction to control for multiple comparisons, identifying specific group differences when the Kruskal–Wallis test indicated significance. A Wilcoxon rank-sum test was used to compare the 18–35-year and 36–50-year groups. Relationships between biomarkers and PSSI-5 domains were assessed using Spearman’s rank correlation coefficient.

#### Characteristics of the Statistical Tool

Analyses were conducted using the R Statistical language (version 4.3.3; [70]) on Windows 11 Pro 64 bit (build 26100) using the packages *Hmisc* (version 5.2.0; [71]), *ggpubr* (version 0.6.0; [72]), *report* (version 0.5.8; [73]), *RColorBrewer* (version 1.1.3; [74]), *gtsummary* (version 1.7.2; [75]), *corrplot* (version 0.94; [76]), *ggplot2* (version 3.5.0; [77]), *stringr* (version 1.5.1; [78]), *dplyr* (version 1.1.4; [79]), and *tidyr* (version 1.3.1; [80]).

## 5. Conclusions

As an overall conclusion, the above findings stress some potential of MMP-2 as a biomarker for monitoring PTSD severity, particularly in the early recovery phase (Past PTSD ≤5 y), where it is linked to overall symptom burden and arousal symptoms. The results of our study point to promising new horizons for the diagnostic biomarkers of PTSD symptoms. It is most probable that such biomarkers will form a panel of biochemical tests and clinical observations, which, when combined, will contribute to increasing the specificity and sensitivity of these diagnostic tools.

## Figures and Tables

**Figure 1 ijms-26-05662-f001:**
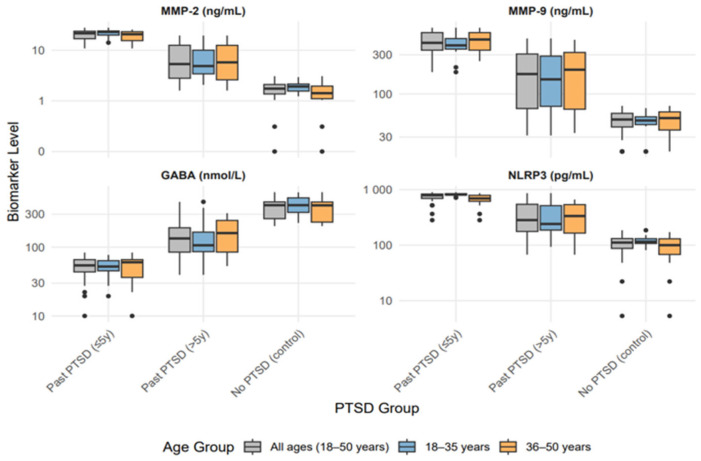
Biomarker levels across PTSD groups by age distribution (MMP-2: Matrix Metalloproteinase-2, MMP-9: Matrix Metalloproteinase-9 GABA: Gamma-Aminobutyric Acid, NLRP3: NOD-Like Receptor Pyrin Domain-Containing Protein 3).

**Figure 2 ijms-26-05662-f002:**
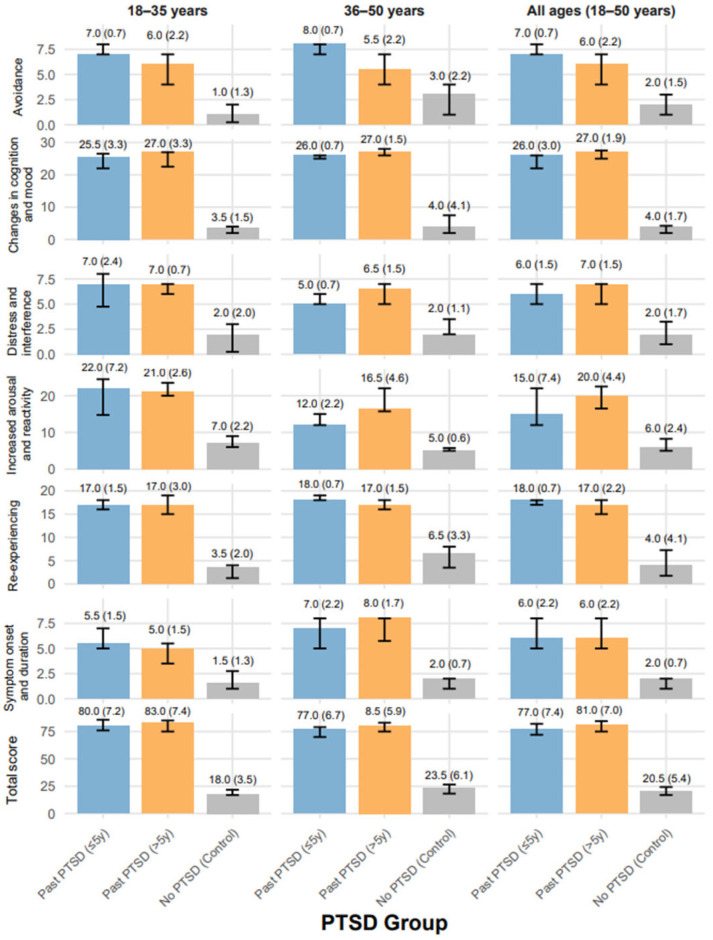
Median PSSI-5 scores with variability (SE) across PTSD domains, groups, and age distributions (PSSI-5_ReExp: PSSI-5 re-experiencing subscale score, PSSI-5_Avoid: PSSI-5 avoidance subscale score, PSSI-5_CogMood: PSSI-5 changes in cognition and mood subscale score, PSSI-5_Arousal: PSSI-5 increased arousal and reactivity subscale score, PSSI-5_Distress: PSSI-5 distress and interference score, PSSI-5_OnsetDur: PSSI-5 symptom onset and duration score, PSSI-5_Total: PSSI-5 total score).

**Figure 3 ijms-26-05662-f003:**
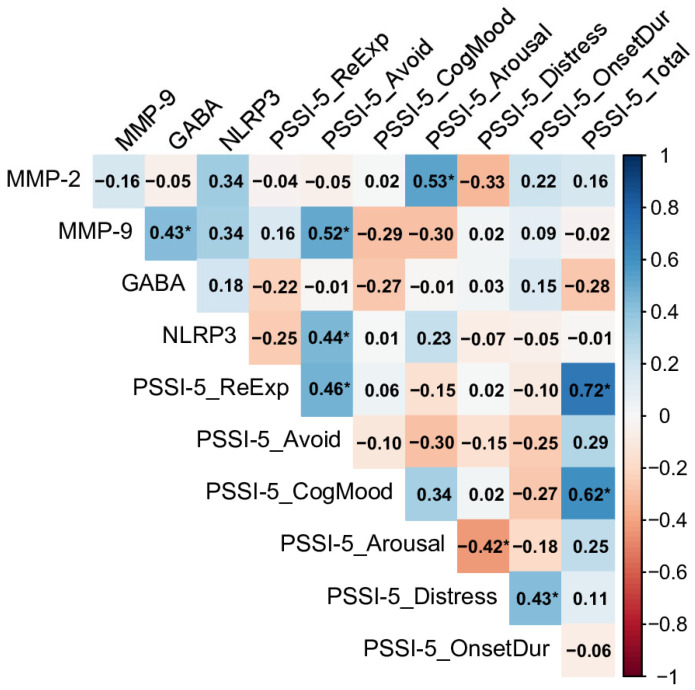
Spearman correlation matrix of biomarkers (MMP-2, MMP-9, GABA, NLRP3) and PSSI-5 scores in individuals without PTSD (n = 28). Significant correlations (*p* < 0.05) are marked with an asterisk (MMP-2: Matrix Metalloproteinase-2, MMP-9: Matrix Metalloproteinase-9 GABA: Gamma-Aminobutyric Acid, NLRP3: NOD-Like Receptor Pyrin Domain-Containing Protein 3, PSSI-5_ReExp: PSSI-5 re-experiencing subscale score, PSSI-5_Avoid: PSSI-5 avoidance subscale score, PSSI-5_CogMood: PSSI-5 changes in cognition and mood subscale score, PSSI-5_Arousal: PSSI-5 increased arousal and reactivity subscale score, PSSI-5_Distress: PSSI-5 distress and interference score, PSSI-5_OnsetDur: PSSI-5 symptom onset and duration score, PSSI-5_Total: PSSI-5 total score).

**Figure 4 ijms-26-05662-f004:**
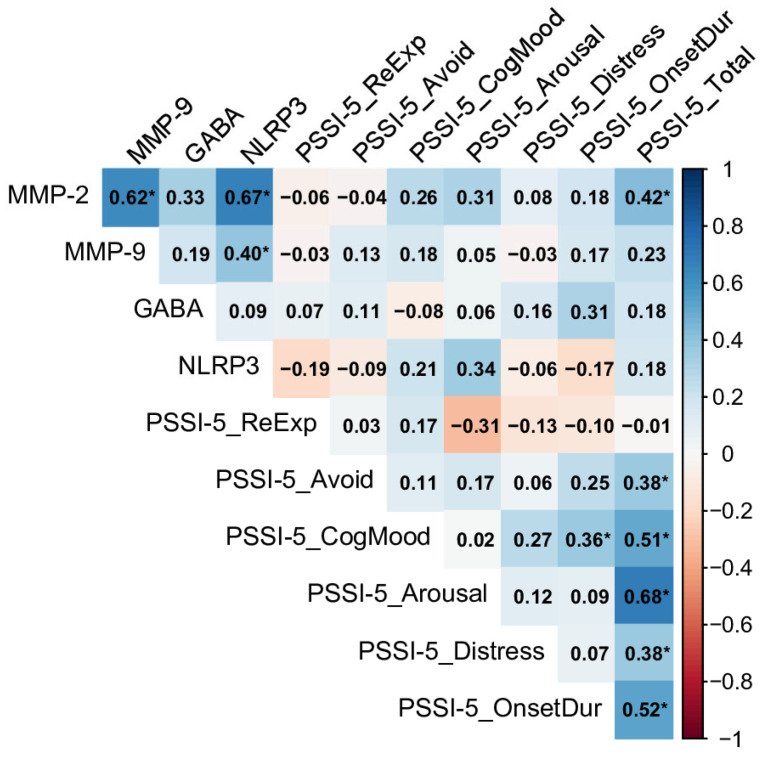
Spearman correlation matrix of biomarkers (MMP-2, MMP-9, GABA, NLRP3) and PSSI-5 scores in individuals with Past PTSD (>5 y), n = 31. Significant correlations (*p* < 0.05) are marked with an asterisk (MMP-2: Matrix Metalloproteinase-2, MMP-9: Matrix Metalloproteinase-9 GABA: Gamma-Aminobutyric Acid, NLRP3: NOD-Like Receptor Pyrin Domain-Containing Protein 3, PSSI-5_ReExp: PSSI-5 re-experiencing subscale score, PSSI-5_Avoid: PSSI-5 avoidance subscale score, PSSI-5_CogMood: PSSI-5 changes in cognition and mood subscale score, PSSI-5_Arousal: PSSI-5 increased arousal and reactivity subscale score, PSSI-5_Distress: PSSI-5 distress and interference score, PSSI-5_OnsetDur: PSSI-5 symptom onset and duration score, PSSI-5_Total: PSSI-5 total score).

**Figure 5 ijms-26-05662-f005:**
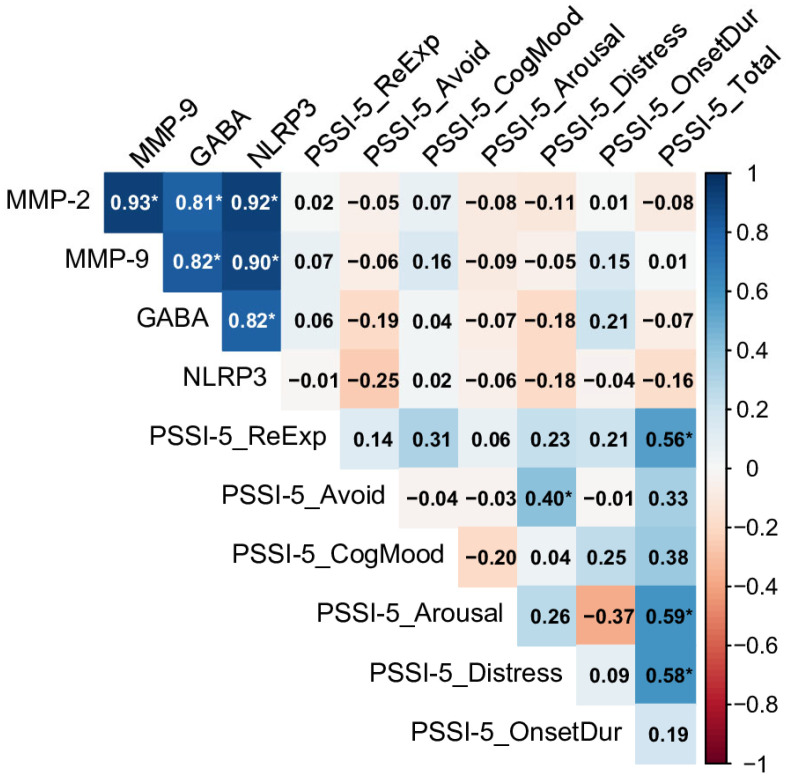
Spearman correlation matrix of biomarkers (MMP-2, MMP-9, GABA, NLRP3) and PSSI-5 scores in individuals with Past PTSD (≤ 5 y), n = 33. Significant correlations (*p* < 0.05) are marked with an asterisk (MMP-2: Matrix Metalloproteinase-2, MMP-9: Matrix Metalloproteinase-9 GABA: Gamma-Aminobutyric Acid, NLRP3: NOD-Like Receptor Pyrin Domain-Containing Protein 3, PSSI-5_ReExp: PSSI-5 re-experiencing subscale score, PSSI-5_Avoid: PSSI-5 avoidance subscale score, PSSI-5_CogMood: PSSI-5 changes in cognition and mood subscale score, PSSI-5_Arousal: PSSI-5 increased arousal and reactivity subscale score, PSSI-5_Distress: PSSI-5 distress and interference score, PSSI-5_OnsetDur: PSSI-5 symptom onset and duration score, PSSI-5_Total: PSSI-5 total score).

**Table 1 ijms-26-05662-t001:** Biomarker levels across PTSD groups by age distribution.

Characteristic	N	Past PTSD (≤5 y) (N = 33)	Past PTSD (>5 y) (N = 31)	No PTSD (Control) (N = 28)	*p*
MMP-2, ng/mL					
All ages (18–50 years)	92	21.66 ^a^ (16.90–23.80)	5.34 ^b^ (2.81–12.55)	1.75 ^c^ (1.37–2.11)	<0.001
18–35 years	45	23.09 ^a^ (19.96–24.03)	4.88 ^b^ (3.42–10.35)	1.92 ^c^ (1.56–2.15)	<0.001
36–50 years	47	20.74 ^a^ (15.48–23.27)	5.78 ^b^ (2.61–12.51)	1.42 ^c^ (1.11–1.96)	<0.001
MMP-9, ng/mL					<0.001
All ages (18–50 years)	92	418.70 ^a^ (341.08–558.19)	175.00 ^b^ (66.64–306.66)	48.99 ^c^ (39.77–58.23)	<0.001
18–35 years	45	390.13 ^a^ (351.56–476.93)	150.73 ^b^ (70.97–289.14)	47.64 ^c^ (42.47–52.88)	<0.001
36–50 years	47	460.07 ^a^ (341.08–558.19)	197.50 ^b^ (65.47–320.99)	51.00 ^c^ (36.51–60.64)	<0.001
GABA, nmol/L					<0.001
All ages (18–50 years)	92	54.33 ^a^ (43.68–65.87)	133.73 ^b^ (84.67–191.29)	406.94 ^c^ (259.67–454.33)	<0.001
18–35 years	45	52.29 ^a^ (45.32–63.68)	106.42 ^b^ (86.42–165.31)	406.94 ^c^ (319.98–521.89)	<0.001
36–50 years	47	60.40 ^a^ (36.30–65.87)	160.14 ^b^ (84.79–245.58)	406.53 ^c^ (230.99–452.47)	<0.001
NLRP3, pg/mL					<0.001
All ages (18–50 years)	92	791.50 ^a^ (693.23–832.14)	282.80 ^b^ (176.51–543.72)	111.41 ^c^ (87.50–131.37)	<0.001
18–35 years	45	810.84 ^a^ (798.15–842.00)	240.74 ^b^ (186.69–544.02)	115.33 ^c^ (107.32–130.73)	<0.001
36–50 years	47	693.23 ^a^ (615.89–788.30)	339.08 ^b^ (164.60–536.33)	100.56 ^c^ (68.02–130.13)	<0.001

Notes: Values are presented as median (interquartile range, Q1–Q3). *p*-values were calculated using the Kruskal–Wallis rank sum test. For characteristics with significant differences (*p* < 0.05), groups sharing the same superscript letter (^a^, ^b^, ^c^) do not differ significantly, while different letters indicate significant differences based on post hoc testing (MMP-2: Matrix Metalloproteinase-2, MMP-9: Matrix Metalloproteinase-9 GABA: Gamma-Aminobutyric Acid, NLRP3: NOD-Like Receptor Pyrin Domain-Containing Protein 3).

**Table 2 ijms-26-05662-t002:** PSSI-5 scores across PTSD groups by age distribution.

Characteristic	N	Past PTSD (≤5 y) (N = 33)	Past PTSD (>5 y) (N = 31)	No PTSD (Control) (N = 28)	*p*
PSSI-5 Re-experiencing					
All ages (18–50 years)	92	18.00 ^a^ (17.00–18.00)	17.00 ^b^ (15.00–18.00)	4.00 ^c^ (1.75–7.25)	<0.001
18–35 years	45	17.00 ^a^ (16.00–18.00)	17.00 ^a^ (15.00–19.00)	3.50 ^b^ (1.25–4.00)	<0.001
36–50 years	47	18.00 ^a^ (18.00–19.00)	17.00 ^a^ (16.00–18.00)	6.50 ^b^ (3.50–8.00)	<0.001
PSSI-5 Avoidance					<0.001
All ages (18–50 years)	92	7.00 ^a^ (7.00–8.00)	6.00 ^b^ (4.00–7.00)	2.00 ^c^ (1.00–3.00)	<0.001
18–35 years	45	7.00 ^a^ (7.00–8.00)	6.00 ^a^ (4.00–7.00)	1.00 ^b^ (0.25–2.00)	<0.001
36–50 years	47	8.00 ^a^ (7.00–8.00)	5.50 ^b^ (4.00–7.00)	3.00 ^c^ (1.00–4.00)	<0.001
PSSI-5 Changes in Cognition and Mood					<0.001
All ages (18–50 years)	92	26.00 ^a^ (22.00–26.00)	27.00 ^a^ (25.00–27.50)	4.00 ^b^ (2.00–4.25)	<0.001
18–35 years	45	25.50 ^a^ (22.00–26.50)	27.00 ^a^ (22.50–27.00)	3.50 ^b^ (2.00–4.00)	<0.001
36–50 years	47	26.00 ^a^ (25.00–26.00)	27.00 ^a^ (26.00–28.00)	4.00 ^b^ (2.00–7.50)	<0.001
PSSI-5 Increased Arousal and Reactivity					<0.001
All ages (18–50 years)	92	15.00 ^a^ (12.00–22.00)	20.00 ^b^ (16.50–22.50)	6.00 ^c^ (5.00–8.25)	<0.001
18–35 years	45	22.00 ^a^ (14.75–24.50)	21.00 ^b^ (20.00–23.50)	7.00 ^b^ (6.00–9.00)	<0.001
36–50 years	47	12.00 ^a^ (12.00–15.00)	16.50 ^b^ (15.75–22.00)	5.00 ^c^ (5.00–5.75)	<0.001
PSSI-5 Distress and Interference					<0.001
All ages (18–50 years)	92	6.00 ^a^ (5.00–7.00)	7.00 ^a^ (5.00–7.00)	2.00 ^b^ (1.00–3.25)	<0.001
18–35 years	45	7.00 ^a^ (4.75–8.00)	7.00 ^a^ (6.00–7.00)	2.00 ^b^ (0.25–3.00)	<0.001
36–50 years	47	5.00 ^a^ (5.00–6.00)	6.50 ^a^ (5.00–7.00)	2.00 ^b^ (2.00–3.50)	<0.001
PSSI-5 Symptom Onset and Duration					<0.001
All ages (18–50 years)	92	6.00 ^a^ (5.00–8.00)	6.00 ^a^ (5.00–8.00)	2.00 ^b^ (1.00–2.00)	<0.001
18–35 years	45	5.50 ^a^ (5.00–7.00)	5.00 ^a^ (3.50–5.50)	1.50 ^b^ (1.00–2.75)	<0.001
36–50 years	47	7.00 ^a^ (5.00–8.00)	8.00 ^a^ (5.75–8.00)	2.00 ^b^ (1.00–2.00)	<0.001
PSSI-5 Total					<0.001
All ages (18–50 years)	92	77.00 ^a^ (72.00–82.00)	81.00 ^a^ (75.00–84.50)	20.50 ^b^ (17.00–24.25)	<0.001
18–35 years	45	80.00 ^a^ (76.00–85.75)	83.00 ^a^ (75.00–85.00)	18.00 ^b^ (17.00–21.75)	<0.001
36–50 years	47	77.00 ^a^ (70.00–79.00)	80.50 ^a^ (75.00–83.00)	23.50 ^b^ (18.25–26.50)	<0.001

Notes: alues are presented as median (interquartile range, Q1–Q3). *p*-values were calculated using the Kruskal–Wallis rank sum test. For characteristics with significant differences (*p* < 0.05), groups sharing the same superscript letter (^a^, ^b^, ^c^) do not differ significantly, while different letters indicate significant differences based on post hoc testing (PSSI-5_ReExp: PSSI-5 re-experiencing subscale score, PSSI-5_Avoid: PSSI-5 avoidance subscale score, PSSI-5_CogMood: PSSI-5 changes in cognition and mood subscale score, PSSI-5_Arousal: PSSI-5 increased arousal and reactivity subscale score, PSSI-5_Distress: PSSI-5 distress and interference score, PSSI-5_OnsetDur: PSSI-5 symptom onset and duration score, PSSI-5_Total: PSSI-5 total score).

**Table 3 ijms-26-05662-t003:** Demographic and clinical characteristics of participants across PTSD groups.

Characteristic	Total (N = 92)	Past PTSD (≤5 y) (N = 33)	Past PTSD (>5 y) (N = 31)	No PTSD (Control) (N = 28)	*p*
Age, years	34.0 (27.0–41.0)	34.0 (31.0–41.0)	36.0 (29.5–41.0)	33.5 (24.3–41.5)	0.524
Employment in hazardous conditions, years	10.0 (6.0–14.0)	11.0 (7.0–14.0)	10.0 (7.5–15.0)	10.0 (3.0–14.0)	0.418
BMI, kg/m^2^	23.0 (22.0–26.0)	22.0 ^a^ (22.0–24.0)	22.0 ^a^ (20.0–24.5)	25.0 ^b^ (22.0–28.3)	0.011
Daily cigarette consumption, cigarettes/day	5.0 (0.0–20.0)	5.0 ^b^ (5.0–20.0)	5.0 ^ab^ (2.5–20.0)	1.0 ^a^ (0.0–6.3)	0.033

Notes: values are presented as median (interquartile range, Q1–Q3). *p*-values were calculated using the Kruskal–Wallis rank sum test. For characteristics with significant differences (*p* < 0.05), groups sharing the same superscript letter (^a^, ^b^) do not differ significantly, while different letters indicate significant differences based on post hoc testing.

## Data Availability

All data and analysis are available within the manuscript, or upon request to the corresponding author.

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
