# Peer review of "Involvement of Matrix Metalloproteinases (MMP-2 and MMP-9), Inflammasome NLRP3, and Gamma-Aminobutyric Acid (GABA) Pathway in Cellular Mechanisms of Neuroinflammation in PTSD"

_ijms, 2025, doi:10.3390/ijms26125662_

Round 1
Reviewer 1 Report
Comments and Suggestions for Authors
This paper examines the link between PTSD and neuroinflammation, focusing on three biomarkers: MMP-2, MMP-9, NLRP3 inflammasome and GABA. The study involves 92 participants divided into three groups: recent PTSD (≤5 years), remote PTSD (>5 years), and a control group without PTSD. Plasma concentrations of MMP-2, MMP-9, NLRP3 and GABA were measured, and symptom severity was assessed using the PSS-I-5. Results showed that recent PTSD group had significantly higher levels of MMP-2, MMP-9 and NLRP3, and lower levels of GABA compared to the remote PTSD and control groups. This suggests a stronger neuroinflammatory response and inhibitory neurotransmitter dysfunction in recent PTSD patients. Moreover, MMP-2 levels were significantly correlated with total PTSD symptom scores, particularly in the recent PTSD group, indicating MMP-2's potential as a biomarker for PTSD severity. The study also found strong correlations among MMP-2, MMP-9 and NLRP3. Reduced GABA levels may disrupt inhibitory signaling, worsening PTSD symptoms. These findings offer new insights into PTSD's pathophysiology and provide potential targets for developing targeted therapies. While many concerns should be addressed before the manuscript can be accepted for publication in <Biology>
Major points:
- NLRP3 is a pivotal molecule in the inflammasome signaling pathway, not a secreted protein. Why can serum NLRP3 detection reflect more intense neuroinflammation? Similarly, IL-1β and IL-18, as secreted proteins resulting from inflammasome activation, are robust serum biomarkers for detecting inflammation. However, the authors did not measure these inflammatory secreted proteins.
- The study found strong correlations between GABA levels and MMP-2, MMP-9, and NLRP3, but it did not sufficiently explore the mechanisms of interaction between GABA and other neuroinflammatory factors. How does more intense neuroinflammation lead to reduced GABA levels and inhibitory neurotransmitter dysfunction? This question needs further analysis in the discussion section.
- This study classified PTSD patients using a 5 - year midpoint, failing to capture dynamic changes over time. A longitudinal design is suggested, with multiple assessment time points for PTSD patients (e.g., 1, 3, 6, and 12 months post-trauma). This would enable analysis of changes in biomarker data and better capture of dynamic changes.
- The study used Spearman correlation analysis to explore the relationship between biomarkers and PTSD symptoms, but this method cannot control for potential confounding factors such as age, gender, comorbidities, and lifestyle. More advanced statistical methods (e.g., multiple linear regression or structural equation modeling) are suggested to analyze the relationship between biomarkers and PTSD symptoms while controlling for confounding factors.
- PTSD patients may exhibit significant heterogeneity in symptom presentation. Failure to distinguish subtypes may mask important findings, such as differentiating avoidance behavior into avoidance of trauma-related stimuli and general stress stimuli. The authors are advised to analyze whether GABA, MMP-2, MMP-9, and NLRP3 levels differ across PTSD subtypes based on existing data.
Minor points:
- In section 4.4 Statistical Analysis, the letter P representing differences should be in italics.
- Greek letters should not be replaced by Latin letters in units, e.g., use μg/dL instead of ug/dl.
3.There appear to be some redundant spaces in certain places, such as line 384.
Author Response
Dear Editors,
In response to Reviewer 1’s comments, the suggested revisions have been made to the manuscript.
- Page 13 – The following statement was added:"NLRP3 is a pivotal molecule in the inflammasome signaling pathway."The authors intend to describe the roles of IL-1β and IL-18 in PTSD in a subsequent publication.
- Linking Neuroinflammation to Reduced GABA Levels and Inhibitory Neurotransmitter Dysfunction (Addition to discussion section)
The significant correlations observed between GABA and neuroinflammatory biomarkers (MMP-2, MMP-9, and NLRP3) in the Past PTSD (>5y) group, alongside markedly reduced GABA levels in both PTSD groups compared to controls, indicate a dynamic relationship between heightened neuroinflammation and GABAergic impairment (Section 2.3, Figure 4; Section 2.1, Table 1). Specifically, the Past PTSD (>5y) group showed a strong correlation between MMP-2 and GABA (rho = 0.81, Section 2.3, Figure 4), while GABA levels were significantly lower in the Past PTSD (≤5y) group (median: 54.33 nmol/L, IQR: 43.68–65.87) compared to controls (median: 406.94 nmol/L, IQR: 259.67–454.33, p < 0.001, Section 2.1, Table 1). Elevated neuroinflammation, characterized by increased MMP-2, MMP-9, and NLRP3 levels, likely contributes to reduced GABA concentrations and compromised inhibitory neurotransmission through several mechanistic pathways.One key mechanism involves the activation of the NLRP3 inflammasome, which triggers the release of pro-inflammatory cytokines such as interleukin-1β (IL-1β) and tumor necrosis factor-α (TNF-α). These cytokines can disrupt GABAergic signaling by modulating GABA_A receptor expression and function. Research has demonstrated that IL-1β reduces GABA_A receptor subunit expression, weakening inhibitory synaptic transmission and promoting neuronal hyperexcitability [1]. In this study, NLRP3 levels were significantly elevated in the Past PTSD (≤5y) group (median: 791.50 pg/mL, IQR: 693.23–832.14) compared to controls (median: 111.41 pg/mL, IQR: 87.50–131.37, p < 0.001, Section 2.1, Table 1), reflecting a potent inflammatory state that likely exacerbates GABAergic deficits, as evidenced by the low GABA levels in this group (Section 2.1, Table 1).Another pathway involves the actions of MMP-2 and MMP-9, which degrade extracellular matrix components, including perineuronal nets (PNNs) that encase GABAergic interneurons. These nets stabilize inhibitory synapses and shield interneurons from inflammatory and oxidative damage. Elevated MMP-2 (median: 21.66 ng/mL, IQR: 16.90–23.80) and MMP-9 (median: 418.70 ng/mL, IQR: 341.08–558.19) levels in the Past PTSD (≤5y) group compared to controls (MMP-2: median: 1.75 ng/mL, IQR: 1.37–2.11; MMP-9: median: 48.99 ng/mL, IQR: 39.77–58.23, p < 0.001, Section 2.1, Table 1) may compromise PNN integrity, impairing GABA synthesis and release. This disruption could account for the observed reduction in GABA levels and diminished inhibitory function. The strong correlation between MMP-2 and GABA in the Past PTSD (>5y) group (rho = 0.81, Section 2.3, Figure 4) further supports the notion that sustained MMP activity perpetuates GABAergic dysfunction over time.Additionally, neuroinflammation fosters oxidative stress, which adversely affects GABAergic interneurons. Both NLRP3 activation and elevated MMP activity generate reactive oxygen species (ROS), which can inhibit glutamic acid decarboxylase (GAD), the enzyme critical for GABA synthesis. This inhibition may explain the markedly reduced GABA levels in the Past PTSD (≤5y) group (median: 54.33 nmol/L, IQR: 43.68–65.87, Section 2.1, Table 1), contributing to an imbalance in inhibitory signaling that exacerbates PTSD symptoms such as increased arousal and reactivity (median score: 15.00 in Past PTSD ≤5y vs. 6.00 in controls, p < 0.001, Section 2.2, Table 2).Chronic neuroinflammation may also disrupt the excitatory-inhibitory balance by promoting glutamate excitotoxicity. High MMP-9 levels, particularly in the Past PTSD (≤5y) group (median: 418.70 ng/mL, IQR: 341.08–558.19, Section 2.1, Table 1), are known to enhance glutamate release and impair astrocytic glutamate uptake, leading to excessive excitatory signaling [2]. This imbalance further suppresses GABAergic activity, reinforcing the observed GABA reductions and potentially intensifying symptoms like hypervigilance and emotional dysregulation (Section 2.2, Table 2). Despite the absence of direct correlations between GABA and PSSI-5 domains across all groups (Section 2.3, Figures 3–5), the significantly lower GABA levels in PTSD groups indicate a broader inhibitory deficit that may amplify the effects of neuroinflammatory mediators. For instance, MMP-2’s significant correlation with the PSSI-5 Total Score in the Past PTSD (≤5y) group (rho = 0.42, Section 2.3, Figure 5) suggests that reduced GABAergic tone may enhance the inflammatory contribution to symptom severity, particularly in early recovery. These findings highlight the intricate relationship between neuroinflammation and GABAergic dysfunction, emphasizing the potential of targeting inflammatory and GABAergic pathways to mitigate PTSD symptoms.
- In this study, patients with PTSD were classified using a 5-year midpoint, as this was the foundational assumption of the entire project. The authors thank the reviewer for the suggestion to conduct longitudinal measurements at different time points. This is an interesting idea that we will consider when planning our next study.
- We thank the reviewer for suggesting advanced statistical methods, such as multiple linear regression or structural equation modeling (SEM), to control for potential confounders like age, gender, comorbidities, and lifestyle in analyzing relationships between biomarkers (MMP-2, MMP-9, GABA, NLRP3) and PTSD symptoms assessed via the PTSD Symptom Scale Interview for DSM-5 (PSSI-5). While these methods have merit, we believe the Spearman correlation analysis used (Section 4.4, Section 2.3) was appropriate for our exploratory objectives and study design, as outlined below. First, Spearman’s rank correlation was chosen to explore associations across multivariate data, suitable for our non-normally distributed data (Section 4.4). It identified key relationships, such as MMP-2 with the PSSI-5 Total Score in Past PTSD (≤5y) (rho = 0.42, Section 2.3, Figure 5) and MMP-2 with GABA in Past PTSD (>5y) (rho = 0.81, Section 2.3, Figure 4), aligning with our aim to highlight all possible value paths for future studies, rather than confirming specific hypotheses.Second, demographic characteristics were reported in Table 3 (Section 4.1). The all-male sample (n = 92) eliminated gender as a confounder, with no significant differences in age (p = 0.524) or employment in hazardous conditions (p = 0.418). Although BMI and cigarette consumption differed (p = 0.011 and p = 0.033, Table 3), these are well-known outcomes of PTSD exposure, driven by stress-related behavioral changes, not independent predictors [1]. Thus, they were not primary confounders for our focus on neuroinflammatory and GABAergic pathways (Section 2.1, Section 2.3). This homogeneity supports our univariate approach.Third, stratification by PTSD duration (≤5y, n = 33; >5y, n = 31; no-PTSD, n = 28, Section 4.1) and age (18–35y, n = 45; 36–50y, n = 47, Section 2.1, Table 1; Section 2.2, Table 2) distinguished profiles, with distinct biomarker levels (e.g., MMP-2: 21.66 ng/mL in Past PTSD ≤5y vs. 5.34 ng/mL in Past PTSD >5y, p < 0.001, Table 1) and PSSI-5 scores (e.g., Total Score: 77.00 in Past PTSD ≤5y vs. 81.00 in Past PTSD >5y, p < 0.001, Table 2). This approach isolated biomarker-symptom associations effectively. Additionally, multiple linear regression and SEM are outside the scope of our study, which intentionally maintains a wide scope to explore multivariate data without confirming specific hypotheses, as this would overly complicate the analysis. In addition, SEM requires large samples (n > 200) [2], which we simply do not have.
- We acknowledge the reviewer’s concern that heterogeneity in PTSD symptom presentation may mask important findings and their suggestion to analyze whether GABA, MMP-2, MMP-9, and NLRP3 levels differ across PTSD subtypes, such as those distinguished by avoidance behaviors. However, we believe this comment overlaps with the previous concern regarding statistical methods and confounder control, and our study design adequately addresses heterogeneity through stratification. The PTSD cohort was stratified into recent (≤5 years, n = 33) and remote (>5 years, n = 31) resolution groups, alongside a no-PTSD control group (n = 28) (Section 4.1). This stratification, combined with age-specific analyses (18–35 years, n = 45; 36–50 years, n = 47, Section 2.1, Table 1; Section 2.2, Table 2), effectively distinguished patient profiles, as evidenced by significant differences in biomarker levels (e.g., MMP-2: 21.66 ng/mL in Past PTSD ≤5y vs. 5.34 ng/mL in Past PTSD >5y, p < 0.001, Table 1) and PSSI-5 scores (e.g., Total Score: 77.00 in Past PTSD ≤5y vs. 81.00 in Past PTSD >5y, p < 0.001, Table 2). This approach sufficiently mitigated heterogeneity in symptom presentation, including avoidance behaviors, without compromising statistical power
- Minor typographical corrections suggested by the reviewer have been made.
Yours faithfully,
Reviewer 2 Report
Comments and Suggestions for Authors
Review comments on ijms-3625018
Journal: International Journal of Molecular Sciences
Manuscript ID: ijms-3625018
Type of manuscript: Article
Title: Involvement of Matrix Metalloproteinases (MMP-2 and MMP-9), Inflammasome NLRP3 and Gamma-Aminobutyric Acid (GABA) Pathway in Molecular Mechanisms of Neuroinflammation in PTSD
Authors: ANNA GRZESIŃSKA *, Ewa Alicja Ogłodek
Molecular Biology
Keywords: gamma-aminobutyric acid; inflammasome NLRP3; neuroinflammation; metalloproteinase; post-traumatic stress disorder
Major comments:
- Title is very misleading since this study does not deal with “MolecularMechanisms of Neuroinflammation in PTSD” but suggest a possible cellular mechanism in neuroinflammation in PTSD.
- Authors stated in the result section that “As an overall conclusion, the above findings stress some potential of MMP-2 as biomarker for monitoring PTSD severity, particularly in the early recovery phase (Past PTSD ≤5y), where they are linked to overall symptom burden and arousal symptoms. (page 11, lines 305-308). As a reviewer, this conclusion seems very important one for the evaluation of this study. Why this or similar description was not included in Abstract or final conclusion section?
- Authors concluded in the end of Discussion section that “In conclusion, these findings underscore the potential of MMP-2 as a biomarker for monitoring PTSD symptom severity, particularly during the early recovery phase (PTSD ≤5 years), where it correlates with overall symptom burden and arousal symptoms” (page 12, lines 403-405). This is very important observation in the present study. Therefore, I wonder why this description does not appear in the very last conclusions and in the abstract section.
- Authors concluded in the very end of the manuscript as “Conclusions” that “the results of our study point to promising new horizons for diagnostic biomarkers of PTSD symptoms. It is most probable that such biomarkers will form a panel of biochemical tests and clinical observations, which, when combined, will contribute to increasing the specificity and sensitivity of these diagnostic tools” (page 14, lines 502-505). This study suggests the importance of several biomarkers for monitoring PTSD symptom severity and this is very important conclusion in the present study. Thus, overall, this study can be possibly publishable in IJMS, but would be more suitable for a clinically-oriented journal.
Minor comments:
- Information on authors is missing in the first page. (page 1)

Author Response
Dear Editors,
In response to Reviewer 2’s comments, the following revisions have been made to the manuscript:
- In accordance with the reviewer’s suggestion, the title of the article has been changed to:Involvement of Matrix Metalloproteinases (MMP-2 and MMP-9), Inflammasome NLRP3 and Gamma-Aminobutyric Acid (GABA) Pathway in Cellular Mechanisms of Neuroinflammation in PTSD
- TheConclusion section in the main body of the article has been expanded.
Yours Faithfully,
The Authors
Round 2
Reviewer 1 Report
Comments and Suggestions for Authors
The revised manuscript has totally addressed my concern, and is suitable for publishing in the journal.
Author Response
Thank you for your valuable comments.
Reviewer 2 Report
Comments and Suggestions for Authors
Review comments on ijms-3625018-revised version
Journal: International Journal of Molecular Sciences
Manuscript ID: ijms-3625018-revised version
Type of manuscript: Article
Title: Involvement of Matrix Metalloproteinases (MMP-2 and MMP-9), Inflammasome NLRP3 and Gamma-Aminobutyric Acid (GABA) Pathway in Cellular Mechanisms of Neuroinflammation in PTSD
Authors: ANNA GRZESIŃSKA *, Ewa Alicja Ogłodek
Molecular Biology
Keywords: gamma-aminobutyric acid; inflammasome NLRP3; neuroinflammation; metalloproteinase; post-traumatic stress disorder
Major comments:
- Title is very misleading since this study does not deal with “MolecularMechanisms of Neuroinflammation in PTSD” but suggest a possible cellular mechanism in neuroinflammation in PTSD.
→ This comment was solved in the revised version by the change of the title.
- Authors stated in the result section that “As an overall conclusion, the above findings stress some potential of MMP-2 as biomarker for monitoring PTSD severity, particularly in the early recovery phase (Past PTSD ≤5y), where they are linked to overall symptom burden and arousal symptoms. (page 11, lines 305-308). As a reviewer, this conclusion seems very important one for the evaluation of this study. Why this or similar description was not included in Abstract or final conclusion section?  
→ Reflecting this comment, authors added some discussions in Discussion section. Further, authors added this part in the conclusion section in the revised version (page 17, lines 572-574).
- Authors concluded in the end of Discussion section that “In conclusion, these findings underscore the potential of MMP-2 as a biomarker for monitoring PTSD symptom severity, particularly during the early recovery phase (PTSD ≤5 years), where it correlates with overall symptom burden and arousal symptoms” (page 12, lines 403-405). This is very important observation in the present study. Therefore, I wonder why this description does not appear in the very last conclusions and in the abstract section.
→ Reflecting this comment, authors added this part in the conclusion section in the revised version (page 17, lines 572-574).
- Authors concluded in the very end of the manuscript as “Conclusions” that “the results of our study point to promising new horizons for diagnostic biomarkers of PTSD symptoms. It is most probable that such biomarkers will form a panel of biochemical tests and clinical observations, which, when combined, will contribute to increasing the specificity and sensitivity of these diagnostic tools” (page 14, lines 502-505). This study suggests the importance of several biomarkers for monitoring PTSD symptom severity and this is very important conclusion in the present study. Thus, overall, this study can be possibly publishable in IJMS, but would be more suitable for a clinically-oriented journal.
→ Although this study is clinically-oriented, I agree that this is still within the scope of IJMS.
Minor comments:
- Information on authors is missing in the first page. (page 1)

Author Response
Thank you for your valuable comments.